# Unraveling the Multifaceted Nexus of Artificial Intelligence Sports and User Willingness: A Focus on Technology Readiness, Perceived Usefulness, and Green Consciousness

**Liqian Gao and Ziyang Liu \***

Graduate School, Kyonggi University, Suwon 16227, Republic of Korea; gaoliqian1581@kyonggi.ac.kr
\* Correspondence: victor@kyonggi.ac.kr

**Abstract:** The adoption of artificial intelligence (AI) in the sports domain, known as AI Sports, has gained considerable attention, while the increasing importance of environmental sustainability further necessitates exploring green-conscious behaviors. This study aims to investigate the impacts of technology readiness, perceived usefulness, and green consciousness on users' willingness to adopt AI Sports. Utilizing a cross-sectional survey and structural equation modeling, data from 670 valid questionnaires were analyzed. The results revealed that technology readiness directly influences users' perceived usefulness of AI Sports, while green awareness significantly affects perceived usefulness, highlighting the relative scarcity of research on green consciousness in this context. Moreover, perceived usefulness plays a crucial role in predicting users' willingness to use AI Sports, contributing to its sustainable development and widespread adoption. This study fills a research gap by exploring the importance of green consciousness in the field of artificial intelligence sports, and provides important insights for the development and promotion of sustainable artificial intelligence sports.

**Keywords:** AI Sports; behavioral intention; green consciousness; perceived usefulness; technology readiness

## 1. Introduction

In recent times, the dynamic synergy between science, technology, and health consciousness has fostered the ascendancy of artificial intelligence sports (AI Sports), an emergent realm that harmoniously amalgamates sports science with cutting-edge artificial intelligence technology. As societal imperatives demand enhanced access to personalized, efficient, and health-oriented fitness solutions, AI Sports has emerged as a compelling response to meet these pressing needs. Within this context, our study endeavors to unravel the multifaceted relationship between AI Sports and its user base, with a primary emphasis on discerning the pivotal roles played by technology readiness, perceived usefulness, and green consciousness in the willingness to use AI Sports.

The confluence of artificial intelligence, big data, and Internet of Things (IoT) technologies has ushered in a new era for AI Sports, yielding vast potential in the realms of health and sports. By harnessing data analytics and personalized guidance, AI Sports not only enhances exercise efficacy and fitness experiences for users but also caters to diverse and individualized exercise requirements, delivering seamless and intelligent exercise solutions. Moreover, this evolutionary trend in AI Sports has spawned novel business prospects, presenting a catalyst for the transformation and innovation of the entire health industry. In parallel, green consciousness, as a prominent steward of environmental consciousness and sustainable values, assumes a paramount role in guiding the trajectory of modern society toward sustainable development. Within the domain of AI Sports, green consciousness significantly influences users' perspectives and attitudes toward AI Sports products and services, as well as their levels of concern for environmental considerations. Therefore, a

comprehensive exploration of the interplay between green consciousness and AI Sports assumes critical importance, as it bears the potential to propel the sustainable advancement of AI Sports, refine product design, and harmonize with the diverse needs of users.

As the prominence of AI Sports escalates, academic inquiry has increasingly explored users' inclination to embrace this innovative domain. However, prevailing studies have predominantly relied on technology acceptance modeling, often overlooking the pivotal role of individual differences in shaping users' willingness to adopt AI Sports. In response, this study aims to elucidate the interplay between technology readiness, perceived usefulness, and green consciousness in influencing users' propensity to utilize AI Sports. Focusing on AI Sports users, this research employs a comprehensive questionnaire survey and data analysis to examine the impact of technology readiness on perceived usefulness, the connection between perceived usefulness and willingness to adopt, and the association of green consciousness with perceived usefulness and willingness to adopt. By discerning these underlying influencing mechanisms, we aim to understand the motivational and decision-making dynamics underlying AI Sports users' behavior, laying a scientific foundation for its further advancement and optimization.

Moreover, this study has multifaceted implications. Firstly, by identifying the determinants of users' willingness to adopt AI Sports, it offers valuable insights to inform product design and marketing strategies in this burgeoning domain. Secondly, the exploration of the correlation between AI Sports and green consciousness provides instructive lessons in integrating AI Sports with the principles of sustainable development, advancing the trajectory of health sports in a sustainable manner. Lastly, this study engenders novel perspectives and broadens the research horizons for academic inquiry in related domains. By shedding light on the intricate relationship between AI Sports and its users, this research should foster informed decision-making and foster advancements within this dynamic landscape.

## 2. Literature Review

### 2.1. AI Sports

AI Sports, which integrates artificial intelligence technology into the realm of sports and health, is dedicated to offering smarter and personalized sports training and monitoring experiences. It focuses on individual fitness and sports training, providing users with scientifically grounded and effective guidance throughout their fitness journey. Lu et al. categorized AI Sports applications into three primary types: wearable devices for body-level detection, monitoring-type devices for preventing sports injuries, and sports equipment for enhancing sports performance [1]. Zheng Fang et al. further subdivided the AI sports industry into intelligent sports manufacturing and intelligent service industries, encompassing the production of sporting goods, construction of intelligent venues, and domains like intelligent sports education, training, and events [2].

Drawing from the classifications proposed by these scholars and aligning them with the research purpose of this study, AI Sports applications can be broadly divided into two categories: (1) applications in daily life and (2) applications in professional events. This investigation primarily centers on AI Sports applications in daily life, which entails the provision of intelligent sports tracking, personalized training plans, intelligent health monitoring, and virtual coaching services through wearable devices and mobile applications. Such applications cater to individual needs and objectives, delivering personalized fitness plans and data monitoring, enabling users to engage in exercise and fitness training with heightened scientific precision.

### 2.2. Technology Readiness

Technology readiness is a pivotal concept used to gauge an individual's acceptance and adoption of novel technological advancements. Parasuraman (2000) emphasizes that comprehending users' personal characteristics and level of technology readiness can facilitate the development of more effective marketing strategies by companies [3]. This understanding, in turn, can foster greater user acceptance and willingness to embrace new

technologies, thus propelling the diffusion and application of cutting-edge innovations. Parasuraman et al. (2001) define technology readiness as an individual's overall propensity to accept new technologies, shaped by a combination of drivers and inhibitors [4].

In research pertaining to technology readiness, scholars commonly analyze this construct through several dimensions, with optimism, innovativeness, discomfort, and insecurity being the most frequently utilized factors. Optimism (Opt) pertains to users' level of positive anticipation toward new technologies or novelties. Higher levels of optimism result in users being more receptive to new technologies and more willing to explore and utilize them. Innovativeness (Inn) refers to users' attitude and desire to innovate when encountering new technologies. Individuals (Ins) with high innovativeness display a more positive outlook toward new technology and are more inclined to explore and apply new functions and features. Discomfort (Dis) reflects the unease experienced by users when using unfamiliar or new technologies. Greater discomfort makes it challenging for users to adapt to new technologies, potentially leading to negative attitudes. Insecurity involves users' feelings of uncertainty stemming from concerns such as privacy breaches and security issues when adopting new technologies. Heightened insecurity makes it difficult for users to place trust in new technologies, possibly leading to negative attitudes [5–8].

In the domain of AI Sports, technology readiness denotes an individual's willingness to embrace AI technology in the context of sports and health. Existing research indicates that individuals with high technology readiness are more inclined to embrace new fitness technology products, adopt AI-assisted personalized fitness training and data monitoring, and adapt to and benefit from fitness experiences driven by new technologies.

### 2.3. Perceived Usefulness

Individuals often develop attitudes and intentions toward learning and using a new technology even before they engage with it. However, these initial attitudes and intentions may not directly or immediately translate into actual use of the technology [9]. A growing number of scholars have recently started to explore the interplay between technology readiness and the technology acceptance model, seeking to integrate these concepts to elucidate individuals' behaviors regarding the adoption of new technologies [10]. Parasuraman et al. have argued that an individual's readiness to use new technology can significantly influence their perception of the technology's usefulness and ease of use, which, in turn, impacts their willingness to adopt and utilize the technology. Building on this, Lin et al. (2005, 2007) amalgamated the Technology Readiness Index (TRI), which encompasses individual characteristics and system-specific dimensions, with the Technology Acceptance Model (TAM) to formulate the Theory Of Technology Readiness and Acceptance Model (TRAM) [11,12]. This comprehensive model elucidates how technology readiness influences users' adoption of new technologies, thereby enhancing its predictive power concerning users' willingness to adopt and use such technologies.

In the context of AI Sports applications, perceived usefulness (PU) pertains to an individual's subjective assessment of the practical value offered by AI technology for fitness training and monitoring. It encompasses users' perceptions of the tangible effects and potential benefits derived from utilizing AI Sports applications. Research has revealed that when users perceive an AI Sports application as capable of delivering effective fitness guidance, improving exercise outcomes, and enhancing overall health, they exhibit a higher likelihood of continued usage and reliance on the technology [13]. This positive perception enhances user satisfaction and fosters loyalty toward the AI Sports application.

### 2.4. Green Consciousness

Green consciousness (GC) refers to an individual's awareness of and concern for environmental issues and sustainable development. It encompasses various aspects, including environmental awareness, attitudes, values, and behaviors. As an increasingly important factor in modern society, green consciousness plays a pivotal role in guiding individuals to take more sustainable and environmentally responsible actions. Research



in the field of consumer decision-making has shown that individuals with higher levels of green consciousness are more likely to prioritize environmental factors when making purchasing decisions [14,15]. They tend to choose products and services that are eco-friendly, sustainable, and consistent with their environmental values. Several studies have explored the relationship between green consciousness and various aspects of consumer decision-making, including product choice, brand loyalty, and willingness to pay a premium for environmentally friendly choices [16]. In the context of sustainable consumption, green consciousness plays a crucial role in driving consumers to use green products and services [17]. Consumers' growing concern about environmental issues and sustainable practices has led to a shift in their preference for eco-friendly products and services. Understanding the interplay between green consciousness and consumer decision-making can help companies develop more targeted and effective sustainability strategies to meet the needs of environmentally conscious consumers [18].

In the field of AI Sports, although relatively little research has been conducted on green consciousness, studying green consciousness not only provides a deeper understanding of user preferences and behaviors but also has implications for product design and marketing strategies. By understanding the interaction between green consciousness and AI Sports, companies can customize their products according to users' environmental preferences and improve the acceptance and satisfaction of environmentally conscious consumers [19]. In addition, the combination of green consciousness and AI Sports highlights the potential to promote sustainable development in the health and sports industries. Adopting eco-friendly practices and aligning AI Sports applications with environmental values contribute to the broader goal of sustainable fitness and well-being [20]. In conclusion, green consciousness is an essential element to consider when examining the dynamics of AI Sports applications. Understanding how environmental awareness influences user perceptions and attitudes toward AI Sports can provide valuable insights for optimizing product design, improving user acceptance, and promoting sustainability in the health and fitness sector. Embracing green consciousness in the development and promotion of AI Sports applications can lead to a more environmentally responsible and sustainable future for the sports and health industry.

*2.5. Behavioral Intention*

Behavioral intention (BI) is a crucial concept in behavioral theory, an individual's readiness to perform a given behavior. It is assumed to be an immediate antecedent of behavior [21]. In 1977, Fishbein and Ajzen introduced the Theory of Reasoned Action (TRA), defining behavioral intention as "the degree of willingness to perform a certain behavior in a specific situation" [22]. TRA posits behavioral intention as the primary predictor of individual behavior. Subsequently, Ajzen expanded TRA in 1985, proposing the Theory of Planned Behavior (TPB), which incorporates individual attitudes, subjective norms, and perceived behavioral control as determinants of behavioral intention [23]. The study of behavioral intentions holds significant importance in predicting and explaining individual behavior and designing effective interventions. Practical applications of this research can foster increased willingness to adopt certain behaviors, thereby facilitating actions to achieve both social and personal objectives. As a result, behavioral intention research finds wide-ranging applications in domains such as consumer behavior, health behavior, and environmental behavior.

## 3. Research Model and Hypotheses

*3.1. Research Model*

Based on the above, the conceptual model of this study integrates technology readiness, perceived usefulness, and green consciousness as key factors influencing user behavior and willingness to use AI Sports. Users with higher technology readiness and positive perceptions of usefulness are more likely to adopt and engage with AI Sports, resulting in improved outcomes. Additionally, the emerging green consciousness in the AI Sports

domain promotes the integration of environmental concepts into sports, contributing to the sustainability of the health industry. Together, these factors create a comprehensive framework for understanding user behavior and acceptance of AI Sports, paving the way for a more sustainable and effective approach to fitness and wellness, as shown in Figure 1 below.

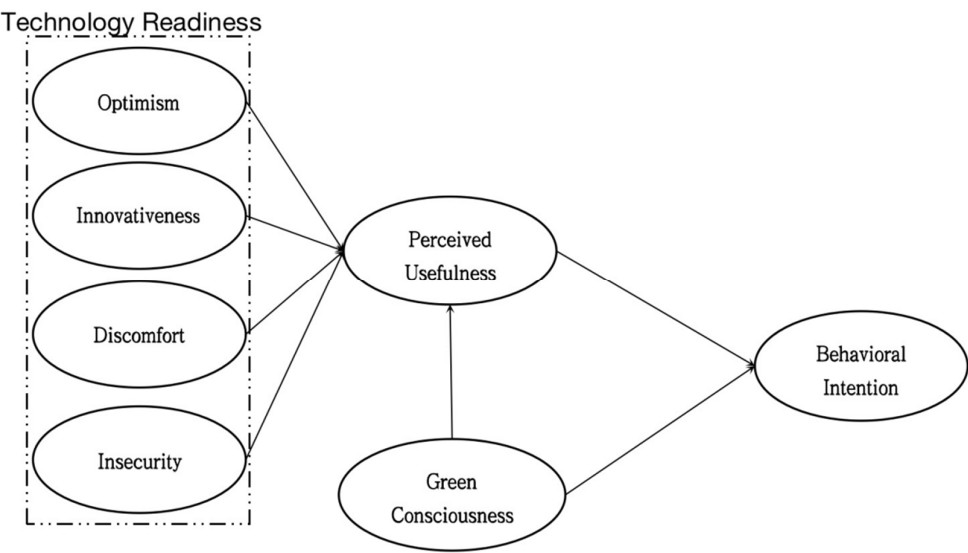

**Figure 1.** Research Conceptual Model Diagram.

### 3.2. The Effect of Technology Readiness on Perceived Usefulness

According to Parasuraman et al.'s research on technology readiness, the most commonly used dimensions were optimism, innovation, discomfort, and insecurity. Optimism is defined as "an individual's positive expectations and beliefs about a new technology, which are manifested in expectations of its potential future advantages and confidence in its successful use". Such optimistic beliefs can foster a positive attitude and adoption of the technology. Gillenson et al. (2002) also discovered a significant positive correlation between optimism and perceived ease of use [24]. This implies that users with optimistic beliefs are more likely to perceive new technologies as easier to use. Their confidence in the future benefits and successful application of the new technology enhances their perception of the technology's usefulness, which, in turn, influences their willingness to adopt it.

These findings offer valuable insights into users' attitudes and behaviors toward AI Sports. In summary, optimism has a positive influence on users' perceived usefulness of AI Sports. Users who hold optimistic beliefs are more likely to perceive AI Sports as helpful for sports and exercise, leading to increased acceptance and willingness to use these. Understanding users' optimism and promoting positive perceptions are crucial for the successful promotion and development of AI Sports. Therefore, the following hypothesis is proposed:

**H1.** *A high level of optimism has a significant positive (+) effect on users' perceived usefulness of adopting AI Sports.*

Innovativeness is defined as "an individual's level of exploration and acceptance of new technologies, encompassing the individual's interest in and desire for the new technology, as well as the individual's positive assessment of the new technology and willingness to use it". Innovativeness plays a crucial role in fostering the adoption and use of new technologies by individuals. Fagan et al. (2012) examined the adoption of virtual reality simulation technology in their study and demonstrated that innovativeness significantly influences the relationships, indicating that higher innovativeness leads to a greater impact of perceived usefulness on user satisfaction and acceptance [25]. These findings suggest that

individuals with greater innovativeness and creativity are more inclined to embrace new technologies and perceive the potential benefits they offer. Highly innovative individuals are more prone to actively explore and experiment with new technologies, holding positive evaluations of the advantages and potential value of such innovations, thus enhancing their perceived usefulness.

Applying these insights to the context of AI Sports, we can hypothesize that users with a high level of innovativeness are more likely to exhibit positive attitudes toward AI Sports. These users develop a keen interest in and desire for the novelty and innovation that AI Sports provides, while also holding positive evaluations of the potential benefits and utility value that AI Sports offers in terms of practical value and personalized sports training. Therefore, there exists a positive association between innovativeness and perceived usefulness. In other words, the higher the level of innovativeness among users, the more likely they are to perceive the potential advantages of AI Sports in enhancing their sports performance and delivering personalized training experiences. As a result, the following hypothesis is proposed:

**H2.** *A high level of innovativeness has a significant positive (+) effect on users' perceived usefulness of adopting AI Sports.*

Discomfort is defined as "the discomfort and concern that individuals feel when confronted with a new technology, which may stem from the individual's unfamiliarity with the new technology, uncertainty, or possible negative consequences". Individuals experiencing discomfort with new technologies tend to feel anxious about using them, which may result in a perceived sense of being controlled by the technology. Kim et al. (2010) examined the use of product virtualization technology in online consumer e-shopping and found that technology anxiety negatively affected users' perceived ease of use and perceived usefulness [26]. Users with technology anxiety faced difficulties in using e-commerce websites and did not find this shopping method very useful. Similarly, Liao and Cheung (2001) observed that technology anxiety had a negative impact on users' perceived ease of use and perceived usefulness [27]. Those who experienced technology anxiety perceived difficulties in Internet shopping and were skeptical about the usefulness of this shopping approach.

In summary, the research suggests that discomfort and technology anxiety can adversely influence individuals' attitudes and perceptions when encountering new technologies. Users who feel discomfort and anxiety are more likely to express skepticism about the practicality and ease of use of new technologies, which may hinder their adoption and use of these technologies. In the context of emerging technology applications like AI Sports, understanding and addressing users' feelings of technology anxiety are critical. By providing targeted support and training to help users overcome their anxiety, AI Sports can enhance perceived usefulness and foster positive willingness to use the technology. Therefore, the following hypothesis is proposed:

**H3.** *A low level of discomfort has a significant positive (+) effect on users' perceived usefulness of adopting AI Sports.*

Insecurity is defined as "individuals' lack of knowledge and mastery of new technologies, feeling that they are unable to cope with the risks and challenges of new technologies, or it may be due to individuals' lack of confidence in the reliability and safety of new technologies". Perceived use of new technologies may pose some risks. Studies conducted by Ronaghi et al. (2020) and Cabanillas et al. (2017) show that cybersecurity issues and users' insecurity negatively affect their attitudes toward adopting new technologies in smart farming and online payment systems, respectively [28,29]. Users express concerns about the security and privacy protection of these technologies, leading to a decrease in their willingness to adopt and use them.

These findings indicate that user insecurity can have a detrimental impact on technology adoption. In the context of emerging technology applications such as AI Sports, user insecurity may hinder users' positive adoption and use of the technology. Addressing users' security and privacy concerns, providing reliable technological safeguards, and boosting users' confidence in the AI Sports are crucial in promoting and developing AI Sports. Therefore, special attention should be given to the issue of user security, and proactive measures should be taken to mitigate user insecurity. By doing so, users' perceived ease of use and perceived usefulness of the AI Sports can be enhanced, leading to a more active adoption and use of the technology. Thus, the following hypothesis is proposed:

**H4.** *A low level of insecurity has a significant positive (+) effect on users' perceived usefulness of adopting AI Sports.*

### 3.3. The Effect of Perceived Usefulness on Behavioral Intention

Perceived usefulness assumes a crucial role in AI Sports, exerting considerable influence on user acceptance and adoption of the technology. It refers to "the extent to which an individual believes that utilizing a specific system would enhance their job performance" [30]. When users perceive a product or service as valuable in addressing their needs and concerns, they are more inclined to utilize it. Davis' (1989) Technology Acceptance Model (TAM) and Venkatesh's (2003) Unified Theory of Acceptance and Use of Technology (UTAUT) provide foundational frameworks for comprehending the significance of perceived usefulness in the adoption of novel technologies [31]. Their research demonstrates that perceived usefulness, alongside perceived ease of use, serves as a prominent predictor of technology adoption and user behavior, positively impacting users' willingness to embrace new technologies. Consequently, in promoting and advancing AI Sports, it is imperative to prioritize enhancing users' perceived usefulness, thereby fostering their ongoing engagement with the technology and contributing to the realization of health-conscious exercise practices and sustainable development. Hence, the following hypothesis is proposed:

**H5.** *A high level of perceived usefulness has a significant positive (+) effect on users' behavioral intention to adopt AI Sports.*

### 3.4. The Effect of Green Consciousness on Perceived Usefulness and Behavioral Intention

Green consciousness, encompassing knowledge, attitudes, values, and behaviors toward environmental issues and sustainable development, plays a vital role in shaping individuals' preferences and decisions. Luzar's (1995) research reveals a positive correlation between green consciousness and environmentally responsible behavior [32]. Moreover, Charter et al. (2017) and Malik et al. (2021) find that green consciousness significantly influences consumption behavior, as individuals with heightened green consciousness exhibit a penchant for eco-friendly and sustainable products, even at premium prices [33,34]. In the realm of AI Sports, green-conscious users are more inclined to opt for environmentally friendly AI Sports products and services. When AI Sports align with the values of green-conscious users by embracing eco-friendly practices, these users are more likely to perceive the practical utility of the application and demonstrate a greater willingness to adopt it. Recognition of an AI Sports application's compatibility with environmental and sustainability objectives may intensify users' focus on its practical outcomes and potential benefits. This optimistic expectation of an eco-friendly AI Sports might further foster users' perception of its usefulness and increase their motivation to engage consistently. Therefore, the following hypotheses are proposed:

**H6.** *A high level of green consciousness has a significant positive (+) effect on users' perceived usefulness of adopting AI Sports.*

**H7.** *A high level of green consciousness has a significant positive (+) effect on users' behavioral intention to adopt AI Sports.*

## 4. Methods

### 4.1. Scale Design and Data Collection

This study employed a cross-sectional survey methodology to explore the interplay between technology readiness, perceived usefulness, and green consciousness on users' behavioral intention to use AI Sports. The cross-sectional survey method captures participants' attitudes, behaviors, and views at a specific point in time, using data collected through an online questionnaire distributed via the "Google Workspace" platform. The participants consisted of a sample of AI Sports users, including current and potential users who have experience with AI Sports. To ensure a representative and accessible sample, convenience sampling was adopted, and participants were invited voluntarily from the "AI & Sports Online Chat Group" based on their eligibility as members. A total of 700 questionnaires were collected, and after conducting the consistency test (KAPPA test), 30 invalid responses were excluded, resulting in 670 valid questionnaires, with a validity rate of 95.71%. The participants' demographic information, such as age, gender, education level, and occupation, was recorded and analyzed to gain insights into the sample composition and characteristics.

To ensure the reliability and validity of the measurement, well-established scales from previous research were employed to assess the variables involved in the study. Minor modifications were made to the measurement items' wording to ensure clarity and appropriateness for the specific context. By investigating and analyzing technology readiness, perceived usefulness, and green consciousness among AI Sports users, this study aims to shed light on their impact on users' behavioral intention to adopt AI Sports. The empirical findings will provide valuable empirical support and scientific guidance for the advancement and growth of AI Sports. The questionnaire encompasses two parts: the first part gathers basic demographic information (demographic analysis characteristics of the data of the valid questionnaire, see Table 1), while the second part consists of measurement items assessing the core variables (including optimism, innovativeness, discomfort, insecurity, green consciousness, perceived usefulness, and behavioral intention).

**Table 1.** Descriptive Statistics of Respondents.

| Statistical Variables | Characteristic | N | Proportion (%) |
|---|---|---|---|
| Gender | Male | 394 | 58.81 |
| | Female | 276 | 41.19 |
| | Total | 670 | 100 |
| Age | Teens | 10 | 1.50 |
| | 20s | 281 | 41.94 |
| | 30s | 248 | 37.01 |
| | 40s | 74 | 11.04 |
| | Over 50 | 57 | 8.51 |
| | Total | 670 | 100 |
| Educational Background | Middle school | 6 | 0.90 |
| | High school | 117 | 17.46 |
| | College | 395 | 58.95 |
| | Graduate school | 152 | 22.69 |
| | Total | 670 | 100 |

The technology readiness measures, including optimism, innovativeness, discomfort, and insecurity, were assessed using Technology Readiness Index (TRI) 2.0 developed by Parasuraman et al. (1997, 2015) [35,36]. This index gauges individuals' propensity to accept and utilize new technologies. Green consciousness (GC) was evaluated using an adapted version of the Green Consciousness Maturity Scale derived from the studies of Mainieri

et al. (1997) and Minton et al. (1997) [37,38]. The scale measures individuals' awareness and concern for environmental issues and sustainable development. To measure perceived usefulness, perceived ease of use, and willingness to use AI Sports, appropriate adaptations of question items from Davis (1989, 1993), Venkatesh et al. (2000), and Adell (2010) were made, considering relevant studies in the field [39–41]. After conducting exploratory factor analysis, 30 variable items in total were finalized for the study. The operational definitions of each variable are presented in Table 2. All measurement items were rated on a 5-point Likert scale, where "1" indicated "strongly disagree", "3" represented a neutral response, and "5" denoted "strongly agree".

**Table 2.** Operational Definition.

| Construct | Operational Definition | N | Reference |
|-----------|------------------------|---|-----------|
| Optimism | Users' positive expectations and beliefs about the application of AI Sports. | 3 | Parasuraman et al. (1997) and (2015) [35,36] |
| Innovativeness | Users' exploration and acceptance of the application of AI Sports. | 3 | |
| Discomfort | Users' discomfort and concern about the application of AI Sports. | 4 | |
| Insecurity | Users' lack of confidence in the reliability and safety of AI Sports. | 4 | |
| Green Consciousness | Users' concern for environmental protection and sustainability in the application of AI Sports. | 5 | Mainieri et al. (1997) and Minton et al. (1997) [37,38] |
| Perceived Usefulness | Users' subjective perception of the actual benefits of applying AI Sports. | 4 | Davis (1989, 1993) and Venkatesh et al. (2000) and Adell (2010) [39–41] |
| Behavioral Intention | Users' willingness to adopt AI Sports. | 3 | |

### 4.2. Data Analysis

Reliability analysis is essential for assessing the consistency and stability of the measurement scales used in the questionnaire. In this study, the Cronbach's alpha coefficient was calculated using the statistical software SPSS 27.0. The customary formula that has been used traditionally is as follows: $\rho_T = \frac{k}{k-1}\left(1 - \frac{\sum_i^k \delta_i^2}{\delta^2 X}\right)$. Generally, a Cronbach's alpha value above 0.9 indicates excellent reliability, between 0.8 and 0.9 indicates good reliability, between 0.7 and 0.8 indicates acceptable reliability, and below 0.7 suggests that the reliability is less satisfactory [42]. The results in Table 3 show that the Cronbach's alpha coefficient value of the model is 0.914, indicating that the survey's reliability is very good. The scale used in the questionnaire exhibits strong internal consistency, meeting the requirements of the study.

**Table 3.** Reliability and Aggregate Validity Analysis.

| Construct | N | Cronbach's Alpha | AVE | CR |
|-----------|---|------------------|-----|-----|
| Optimism | 3 | 0.885 | 0.727 | 0.888 |
| Innovativeness | 3 | 0.853 | 0.674 | 0.86 |
| Discomfort | 4 | 0.874 | 0.642 | 0.877 |
| Insecurity | 4 | 0.876 | 0.642 | 0.877 |
| Perceived Usefulness | 4 | 0.900 | 0.649 | 0.88 |
| Green Consciousness | 5 | 0.875 | 0.651 | 0.902 |
| Behavioral Intention | 5 | 0.890 | 0.628 | 0.893 |
| TOTAL | 28 | 0.914 | / | / |

Validity analysis focuses on the appropriateness and correctness of the questionnaire's design. In this study, validity analysis is carried out through exploratory factor analysis,

comparing the factor loading coefficients of the items to ensure optimal performance for the same factor. Using the statistical software Amos 27.0, composite reliability (CR) and average variance extracted (AVE) are employed to assess the model's convergent validity. The results in Table 3 demonstrate that the CR value for each latent variable exceeds 0.7, indicating good convergent validity for the model. Additionally, as the results of KMO (Kaiser–Meyer–Olkin) test in Table 4 shows that the value of KMO is 0.817, while the results of Bartlett's spherical test show that the significance *p*-value is 0.000 ***, which presents significance at the 1% level, rejecting the original hypothesis that there is a correlation between the variables, and the factor analysis is valid to the extent that it is suitable. And a description of the KMO test is as follows: $\text{KMO} = \frac{\sum\sum_{i \neq j} r_{ij}^2}{\sum\sum_{i \neq j} r_{ij}^2 + \sum\sum_{i \neq j} r_{ij \cdot 1,2,\cdots k}^2}$. Overall, the validity of each latent variable is sound, and the questionnaire demonstrates high quality.

**Table 4.** KMO Test and Bartlett's Test.

| Kaiser–Meyer–Olkin Measure of Sampling Adequacy | | 0.817 |
|---|---|---|
| Bartlett's Test of Sphericity | Approx. Chi-Square | 914.455 |
| | df | 21 |
| | Sig. | 0.000 *** |

*** represents 1% significance levels.

To further evaluate the discriminant validity of each latent variable, Pearson correlation analysis was conducted, and the results are presented in Table 5. The correlation coefficient between each variable and other variables is consistently less than the square root of AVE, indicating good discriminant validity in the questionnaire. These findings collectively support the reliability and validity of the measurement scales, ensuring the accuracy and effectiveness of the data collected in the study.

**Table 5.** Pearson Correlation with AVE Square Root Value.

| | Opt | Inn | Dis | Ins | GC | PU | BI |
|---|---|---|---|---|---|---|---|
| Opt | 0.853 | - | - | - | - | - | - |
| Inn | 0.23 *** | 0.821 | - | - | - | - | - |
| Dis | 0.189 *** | 0.172 *** | 0.801 | - | - | - | - |
| Ins | 0.119 *** | 0.103 *** | 0.139 *** | 0.801 | - | - | - |
| GC | 0.273 *** | 0.245 *** | 0.357 *** | 0.268 *** | 0.807 | - | - |
| PU | 0.319 *** | 0.3 *** | 0.336 *** | 0.261 *** | 0.481 *** | 0.806 | - |
| BI | 0.329 *** | 0.418 *** | 0.327 *** | 0.367 *** | 0.4390 *** | 0.51 *** | 0.792 |

*** represents 1% significance levels; the diagonal numbers are the root values of the AVE for that factor.

Table 3 demonstrates the results of the Cronbach's alpha coefficient, AVE (average variance extracted), and CR (composite reliability) metrics of the model. The Cronbach's alpha coefficient value evaluates whether the collected data are true and reliable, according to which the questions are ranked as unreasonable or haphazardly answered. N (number) is the number of variables involved in the calculation of the reliability analysis.

Table 4 presents the results of the KMO test and Bartlett's sphericity test, which were used to analyze whether factor analysis could be performed. If it passes the KMO test (KMO > 0.6), it shows that there is a correlation between the question variables and meets the requirements of factor analysis. If it passes the Bartlett's test, $p < 0.05$, which is significant, meaning that factor analysis can be performed.

Table 5 shows the parameter results of the model test including the correlation coefficient, significance *p*-value [43].

### 4.3. Structural Equation Model Testing

The results of the reliability and validity factor test have confirmed the suitability of this study for analysis using structural equation modeling (SEM). SEM is a statistical method based on factor analysis and linear regression, used to analyze the complex relationships between multiple variables. It involves downsizing the scale through factor analysis, where multiple variables are converted into one principal component, followed by path analysis to examine the relationships among the variables. In this study, the data from 670 valid questionnaires were analyzed using the statistical software Amos 27.0 to perform SEM. The model fitting process was conducted, and the results are presented in Table 6. The model fitting indexes were all within the reference range, indicating that the model fit was considered good and acceptable.

**Table 6.** Model Fit Index (Statistics).

| Items | $\chi^2$ | df | P | CMIN/DF | GFI | RMSEA | RMR | CFI | NFI | NNFI |
|-------|----------|------|-------|---------|------|-------|------|------|------|------|
| Ideal Value | - | - | >0.05 | <3 | >0.9 | <0.10 | <0.05 | >0.9 | >0.9 | >0.9 |
| Results | 486.321 | 333.000 | 0.000 *** | 1.460 | 0.957 | 0.026 | 0.398 | 0.986 | 0.957 | 0.984 |

*** represents 1% significance levels.

SEM is a powerful analytical tool that allows researchers to explore the complex inter-relationships among variables and test various hypotheses simultaneously. By employing SEM in this study, we can gain deeper insights into the underlying structure and dynamics of the factors influencing users' willingness to use AI Sports applications, and the results obtained are reliable and well supported by the data. The successful application of SEM in this study enhances the robustness and credibility of the findings, contributing to the advancement of knowledge in the field of AI Sports and its implications for sustainable development in the health and sports industries.

Table 6 presents the model fit metrics, commonly used metrics include the chi-square degrees of freedom ratio, GFI, RMSEA, RMR, CFI, NFI, and NNFI. CMIN/DF (chi-square and degrees of freedom df) is mainly used to compare multiple models (the smaller the chi-square value, the better); the degrees of freedom reflect the complexity of the model (the simpler the model, the more degrees of freedom), and vice versa (the more complex the model, the fewer degrees of freedom). The GFI (Goodness-of-Fit Index) mainly applies the coefficient of determination and regression standard deviation to test the fit of the model to the sample observations. Its value is between 0 and 1; the closer to 0 means the worse the fit. A GFI ≥ 0.9 is considered a good model fit. In general, the RMSEA (Root Mean Square Error of Approximation) is below 0.08 (the smaller, the better). The RMR (Root Mean Square Residual) metric measures how well the model fits by measuring the average residual between the predicted correlation and the actual observed correlation. If the RMR is <0.1, the model is considered to be a good fit. The CFI (Comparative Fit Index) has a value between 0 and 1 when comparing hypothetical and independent models, with a value closer to 0 indicating a worse fit and a value closer to 1 indicating a better fit. In general, a CFI ≥ 0.9 is considered a good model fit. For the NNFI (non-normalized fit coefficient) and NFI (normalized fit index), the larger their value, the better the fitted model performs [44].

Table 7 shows the regression coefficients for the path nodes, which can be interpreted as a least-squares one-way linear regression, and it is usually only necessary to look at the *p*-values versus the normalized path coefficients to determine if there is a direct linear influence on that path (X ≥ Y). The impact coefficient mentioned in the text should be clarified with proper citations, and the equation is as follows: $r(X, Y) = \frac{Cov(X,Y)}{\sqrt{Var[X]Var[Y]}}$. One analyzes (*p* < 0.05) whether there is an influence relationship between the model variables based on a significance test. If significance exists, it indicates that there is an influencing

relationship between the variables, which can be analyzed in depth by standardized path coefficients in terms of the amount of influence efficiency [45].

**Table 7.** Table of Model Regression Coefficients.

| | Path | | Unstandardized Coefficient | Standardized Coefficient | Standard Error | Z | P |
|---|---|---|---|---|---|---|---|
| Opt | $\rightarrow$ | PU | 0.157 | 0.163 | 0.036 | 4.360 | 0.000 *** |
| Inn | $\rightarrow$ | PU | 0.160 | 0.158 | 0.038 | 4.174 | 0.000 *** |
| Dis | $\rightarrow$ | PU | 0.172 | 0.170 | 0.039 | 4.363 | 0.000 *** |
| Ins | $\rightarrow$ | PU | 0.130 | 0.123 | 0.040 | 3.270 | 0.001 *** |
| GC | $\rightarrow$ | PU | 0.330 | 0.335 | 0.042 | 7.899 | 0.000 *** |
| PU | $\rightarrow$ | BI | 0.446 | 0.446 | 0.044 | 10.152 | 0.000 *** |
| GC | $\rightarrow$ | BI | 0.240 | 0.243 | 0.042 | 5.691 | 0.000 *** |

*** represents 1% significance levels.

Table 7 shows the following:

- Optimism $\geq$ Perceived usefulness, with a significance *p*-value of 0.000 ***, presenting significance at the 1% level, the original hypothesis is rejected; therefore, this path is valid and its impact coefficient is 0.163.
- Innovativeness $\geq$ Perceived usefulness, with a significance *p*-value of 0.000 *** and presenting significance at the 1% level, the original hypothesis is rejected; therefore, this path is valid and its impact coefficient is 0.158.
- Discomfort $\geq$ Perceived usefulness, with a significance *p*-value of 0.000 *** and presenting significance at the 1% level, the original hypothesis is rejected; therefore, this path is valid and its impact coefficient is 0.17.
- Insecurity $\geq$ Perceived usefulness, with a significance *p*-value of 0.001 *** and presenting significance at the 1% level, the original hypothesis is rejected; therefore, this path is valid and its impact coefficient is 0.123.
- Green consciousness $\geq$ Perceived usefulness, with a significance *p*-value of 0.000 *** and presenting significance at the 1% level, the original hypothesis is rejected; therefore, this path is valid and its impact coefficient is 0.335.
- Perceived usefulness $\geq$ Behavioral intention, with a significance *p*-value of 0.000 *** and presenting significance at the 1% level, the original hypothesis is rejected; therefore, this path is valid and its impact coefficient is 0.446.
- Green consciousness $\geq$ Behavioral intention, with a significance *p*-value of 0.000 *** and presenting significance at the 1% level, the original hypothesis is rejected; therefore, this path is valid and its impact coefficient is 0.243.

## 5. Results and Discussion

### 5.1. Results of the Study

This study uses structural equation modeling (SEM) to analyze the effects of technology readiness, perceived usefulness, and green consciousness on the intention to use AI Sports. In the process of statistical analysis, we collected 670 valid questionnaire data, used the statistical software SPSS 27.0 for questionnaire analysis, and used the statistical software Amos 27.0 for model fitting. The results show that the scale samples in the questionnaire are credible and reliable, and the item design is reasonable; all the indicators of model fitting are within the reference value range, indicating that the fit of the model is good and acceptable. When analyzing the relationship and degree of influence among variables, all paired items showed extremely significant statistical significance ($p < 0.001$). All the assumptions of this study are fully supported, verifying the rationality and effectiveness of the model. Specifically, the impact coefficients for each paired term are as shown in Figure 2. The path from green consciousness to perceived usefulness has the highest impact coefficient of 0.335, indicating a strong influence of green consciousness on perceived usefulness. The path from perceived usefulness to behavioral intention has an impact

coefficient of 0.446, highlighting the significant role of perceived usefulness in predicting behavioral intention. The path from green consciousness to behavioral intention has an impact coefficient of 0.243, showing the substantial impact of green consciousness on users' behavioral intentions.

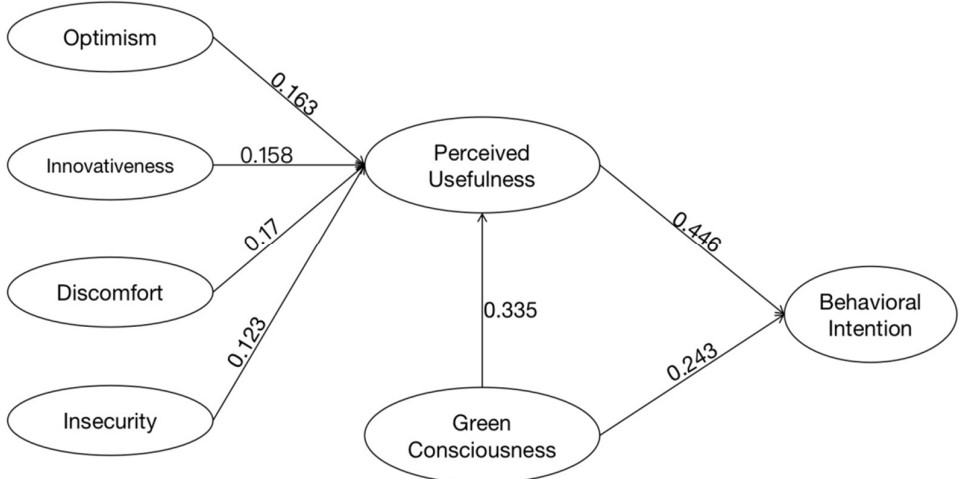

**Figure 2.** Structural Equation Modeling Path Diagram.

*5.2. Discussion*

Based on the results of this study, perceived usefulness also plays a key role in predicting user behavior intentions, which is consistent with previous findings. As users become more aware of the specific benefits of AI Sports, they are more likely to use the technology and apply it to their daily exercise and health management routines. By increasing the perceived usefulness of AI Sports, users are more likely to accept them, thereby encouraging further development and popularization.

Furthermore, green consciousness plays an important and unique role in influencing users' willingness to use AI Sports. Users' green consciousness directly affects their perceived usefulness of AI Sports, and this association is relatively rare in previous studies. There is a dearth of research on green consciousness in the field of AI Sports; however, the results of this study highlight its importance and necessity. Users' consciousness of environmental concerns and their concern for sustainable development plays a significant role in their intentions to use AI Sports. Users with high green consciousness are more inclined to choose AI Sports products or services that are consistent with environmental protection values, so it is easier to recognize the actual value and potential advantages of AI Sports. It is possible that this expectation of being environmentally friendly may lead users to perceive the app as useful and use it more frequently. This discovery is critical to promoting the sustainable development of AI Sports in the field of health and exercise.

In addition, for the promotion and development of AI Sports applications, enterprises and developers can take measures to improve users' technical readiness and encourage them to maintain a positive attitude and optimistic expectations for new technologies. At the same time, attention should be paid to users' green awareness, and the concept of environmental protection should be integrated into the design and marketing strategy of the AI Sports application to meet the needs of users with high environmental awareness for environmentally friendly products, thereby enhancing users' perception of the practicality of the application and their willingness to use it. This study also reminds companies and developers to pay attention to users' discomfort and insecurity. For those users who are uncomfortable with new technologies or worried about safety, corresponding support, and training should be provided to help them overcome their insecurity, enhance their confidence in AI Sports applications, and improve their perception of the practicality of the application and their willingness to use it.

Overall, the findings of this study highlight the scarcity and importance of green consciousness research in the field of AI Sports. An in-depth understanding of users' concerns about environmental issues and consciousness of sustainable development is crucial to the acceptance and willingness to use AI Sports. Therefore, in the promotion and development of AI Sports, more attention should be paid to the role of green consciousness, so as to promote AI Sports in the direction of sustainable and environmentally friendly development. Future research can further explore the influence mechanism of green consciousness in the field of health and sports, so as to better guide user preferences and behaviors, and promote the sustainable development and social benefits of AI Sports.

## 6. Conclusions

This study investigated the influence of technology readiness, perceived usefulness, and green awareness on AI Sports usage willingness through structural equation modeling (SEM). Key findings include the following:

- Technology readiness significantly boosts perceived usefulness, underlining its role in enhancing users' acceptance of AI Sports.
- Green awareness has a substantial impact on AI Sports adoption, highlighting the need for further research in this area.
- Perceived usefulness plays a pivotal role in motivating AI Sports usage, fostering its acceptance and growth.

These factors collectively shape users' willingness to embrace AI Sports. To promote AI Sports applications effectively, a dual focus on environmental awareness and perceived benefits is vital. Future research can delve deeper into the interplay between green consciousness and user preferences in health and exercise technology, contributing to sustainable AI Sports development.

However, this study has certain limitations. First, the convenience sampling method that was employed potentially limits the representativeness of the sample. Therefore, future research could employ more rigorous sampling methods to obtain more representative samples. Secondly, this study mainly focused on the effects of technology readiness, perceived usefulness, and green awareness on users' willingness to use AI Sports, but it did not encompass other potential factors that may influence user behavior. Future research could explore other potential influencing factors to gain a comprehensive understanding of the formation mechanism of user behavior intentions.

This study provides important theoretical and practical implications for understanding AI Sports user behavior and addresses the research gap concerning the role of green awareness in the AI Sports domain. The findings offer valuable insights to promote the sustainable development and environmental friendliness of AI Sports. It is hoped that the results of this study can serve as a beneficial reference for the development and promotion of health and exercise technology applications.

**Author Contributions:** Z.L. designed the study and simulation; L.G. conducted the data analysis; L.G. provided the mathematical methods; L.G. and Z.L. drafted the paper; L.G. and Z.L. edited the paper. All authors have read and agreed to the published version of the manuscript.

**Funding:** This work was supported by Kyonggi University Research Grant 2022.

**Institutional Review Board Statement:** Not applicable.

**Informed Consent Statement:** Not applicable.

**Data Availability Statement:** This study did not report any data.

**Conflicts of Interest:** The authors declare no conflict of interest.

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
