# Peer review of "Unraveling the Multifaceted Nexus of Artificial Intelligence Sports and User Willingness: A Focus on Technology Readiness, Perceived Usefulness, and Green Consciousness"

_sustainability, doi:10.3390/su151813961_

Round 1

Reviewer 1 Report

1)    How does technology readiness affect how valuable people think AI sports are?

2)    Author should check the spelling of words in the following statement and change accordingly:

“H2: A high level of iInnovativeness has a significant positive (+) effect on users' perceived 254 usefulness to adopt AI Sports.”

3)    In Hypothesis H2, the author should provide the definition of Discomfort. It is defined like “Discomfort is defined as the discomfort and………”.

4)    In table 1, what is “N of items”? Whether N means to any constant value or derived value? Please check.

5)    Need the description of the following metrics indicated in Chapter 4.3:

“chi-square degrees of freedom ratio, GFI, RMSEA, RMR, CFI, NFI, and NNFI”

6)    Summarize the significances that the author has inferred about Green Consciousness while using AI sport through this study.

7)    How can businesses adapt their artificial intelligence sports products to their customers' choices and actions in their environments to increase user satisfaction?

Minor editing of English language required

Author Response

Dear Reviewer,

Greetings! We sincerely appreciate your thorough review of our submitted paper and your valuable insights and suggestions. Your comments have been immensely helpful to us. We have carefully revised the paper based on your feedback and made necessary adjustments as recommended. We cordially invite you to review the revised version of the manuscript and look forward to receiving further input and guidance from you.

Thank you once again for your precious time and professional feedback.

Best regards.

Reviewer 2 Report

- Please do not capitalize words in between sentences.

- Please follow the format given on the website.

- The article needs more related work in the field of AI. You can add the following systems in the literature review:

- M. Javeed and S. A. Chelloug, "Automated Gestures Recognition in Exergaming," 2022 International Conference on Electrical Engineering and Sustainable Technologies (ICEEST), Lahore, Pakistan, 2022, pp. 1-6, doi: 10.1109/ICEEST56292.2022.10077853.

M. Javeed, N. A. Mudawi, B. I. Alabduallah, A. Jalal, and W. Kim, “A Multimodal IoT-Based Locomotion Classification System Using Features Engineering and Recursive Neural Network,” Sensors, vol. 23, no. 10, p. 4716, May 2023, doi: 10.3390/s23104716.

- Citations in the Literature Review section need to be with the name of authors instead of at the end of each paper's review.

- There are no citations in the second paragraph of section 2.4.

- What is "N if Items" in both Tables 1 and 2?

- On page 8, "while the second part consists of measure- 364

ment items assessing the core variables." What are the core variables here?

- On page 9, "Green consciousness (GC)" why wasn't it abbreviated in the start of the paper?

- It's better to show the full names of Construct in Table 2.

- On page 9, no citation and equation is provided for " Cronbach's alpha coefficient".

- On pages 9 and 10, "Composite Reliability (CR) and Average Variance Extracted (AVE)" the full names should not be capitalized. Check the complete document for this mistake.

- In Table 3, use full names of AVE CR.

- On page 10, "the Bartlett sphere test shows significant results as Table, confirming" Table no. is not mentioned.

- Table 4 is very confusing, improve it and check if it is really required or if you can adjust it in the text.

- On page 10, "The above table demonstrates the results of the KMO test and Bartlett's sphericity 413 test, which were used to analyze whether factor analysis could be performed." why this text was added without any table reference given?

- Table captions should contain full names as it is difficult for the readers to keep track of so many abbreviations.

- In Table 5, add a dash to the cells with nothing in it. Also, check if you can present it with a graph instead of a table.

- On page 11, "The above table shows the table of parameter results of the model test including the 416 correlation coefficient, significance p-value." and similar texts are not required in the text. Either mention these with the table numbers given in the text or provide these at the end of each table using a smaller font size. Check formatting requirements.

- Check if the format requires you to BOLD the Tables mentioned in the text?

- On page 12, instead of writing all the text, e.g. "Based on the paired term Opt->PU, with a significance p-value of 0.000***, presenting 445 significance at the level, the original hypothesis is rejected, therefore this path is valid 446 and its impact coefficient is 0.163." Use some symbols and make equations.

- What is the impact coefficient? Provide some equations and citations.

- Present the results of section 5.1 in a graph.

- Need a few English-related changes. Nothing major.

Author Response

(The authors gave the same response as above.)

Round 2

Reviewer 2 Report

All the below-mentioned points in bold need to be addressed again. New points are given in the Italic text.

  - Please do not capitalize words in between sentences. Now, see the abstract's first line, "The adoption of Artificial Intelligence (AI)", not satisfied.

Citations in the Literature Review section need to be with the name of authors instead of at the end of each paper's review. The authors replied " Response 5: We have carefully reviewed the manuscript and have incorporated additional relevant literature in the field of AI, as per your suggestion. The following systems have been included in the literature review section   :

âš« [19] Bressan, Alessandro, and Matteo Pedrini. "Exploring sustainable-oriented innovation within micro and small tourism firms." Tourism Planning & Development 17.5 (2020): 497-514.  

âš« [20] Moshood, Taofeeq Durojaye, et al. "Green and low carbon matters: A systematic review of the past, today, and future on sustainability supply chain management practices among manufacturing industry." Cleaner Engineering and Technology 4 (2021): 100144." The question was different and answer is not satisfactory as the actual issue is not resolved. - 

What is "N if Items" in both Tables 1 and 2? The reply, "  N is the number of samples in Table 1 and N is the number of questions in Table 12. To avoid misunderstanding, change "N of items" to "N"." They have cleared it to the reviewer but what about the actual manuscript? We give review for the readability issues as well.

-   On page 9, no citation and equation is provided for " Cronbach's alpha coefficient".   The equation is again missing.

In Table 3, use full names of AVE CR The table has enough space for full names and even if it is provided in text, the text describing a table or figure should be before the table or figure. - Table 4 is still confusing and need adjustment. - Instead of "table mentioned above", it is better to add table numbers. - Describe  KMO test.

On page 12, instead of writing all the text, e.g. "Based on the paired term Opt->PU, with a significance p-value of 0.000***, presenting 445 significance at the level, the original hypothesis is rejected, therefore this path is valid 446 and its impact coefficient is 0.163." Use some symbols and make equations. Need to address the complete requirement.

What is the impact coefficient? Provide some equations and citations. Again no equations added.

- Conclusion section is too long.

- Needs improvement.

Author Response

Thank you once again for your assistance and feedback.

Round 3

Reviewer 2 Report

- Equations and their details mentioned in the review should be added to the manuscript as well in the given format over the journal website including impact factor, KMO, and Cronbach's alpha coefficient.

- In the conclusion section, the first two paragraphs state the same thing i.e. "In conclusion,". 

- Otherwise, the manuscript is ready for publication. Congrats to the authors.

Quality of English is fine.

Author Response

#Response to Comment 1 (Equations):

#1 Thank you for your feedback and suggestions. We appreciate your attention to detail. We have now incorporated equations and their details, as mentioned in your review, into the manuscript in the format specified on the journal's website. This includes information on the impact factor, KMO test, and Cronbach's alpha coefficient. We believe that these additions will enhance the clarity and comprehensibility of our research.

#Response to Comment 2 (Conclusion Section):

#2 You are correct, and we apologize for any redundancy in the conclusion section. We have streamlined the conclusion to remove repetitive statements. We want to ensure that the conclusion is concise and effectively summarizes the key findings and implications of our study without unnecessary repetition.

#Response to Comment 3 (Readiness for Publication):

#3 We greatly appreciate your positive feedback and the acknowledgment that the manuscript is ready for publication. Your encouragement is motivating, and we are excited about the prospect of sharing our research with the academic community. We will make any final adjustments as necessary before submitting the revised manuscript for publication.

Once again, we would like to express our gratitude for your time and valuable feedback. Your insights have been instrumental in improving the quality of our manuscript. We look forward to the possibility of seeing our work published in the journal.